# Inhibitors of the *Plasmodium falciparum* Hsp90 towards Selective Antimalarial Drug Design: The Past, Present and Future

**DOI:** 10.3390/cells10112849

**Published:** 2021-10-22

**Authors:** Melissa Louise Stofberg, Celine Caillet, Marianne de Villiers, Tawanda Zininga

**Affiliations:** Department of Biochemistry, Stellenbosch University, Stellenbosch 7600, South Africa; 18592538@sun.ac.za (M.L.S.); 20010494@sun.ac.za (C.C.); mdevilliers@sun.ac.za (M.d.V.)

**Keywords:** malaria, *Plasmodium falciparum*, molecular chaperone, heat shock protein 90, Hsp90 inhibitors, parasite Hsp90

## Abstract

Malaria is still one of the major killer parasitic diseases in tropical settings, posing a public health threat. The development of antimalarial drug resistance is reversing the gains made in attempts to control the disease. The parasite leads a complex life cycle that has adapted to outwit almost all known antimalarial drugs to date, including the first line of treatment, artesunate. There is a high unmet need to develop new strategies and identify novel therapeutics to reverse antimalarial drug resistance development. Among the strategies, here we focus and discuss the merits of the development of antimalarials targeting the Heat shock protein 90 (Hsp90) due to the central role it plays in protein quality control.

## 1. Introduction

Malaria still poses a major public health burden globally, with an estimated 229 million cases and more than 419,000 deaths; most of these deaths being in the tropics in the year 2019 [1]. Sub-Saharan Africa accounts for about 95% of all malarial deaths, and of these, 67% are children under the age of 5 years [1]. The protozoan parasite responsible for malaria in humans are the single-celled eukaryote *Plasmodium* species in the phylum Apicomplexa. *P. falciparum* is responsible for the deadliest form of the disease in humans [2]. Efforts to eradicate malaria have been limited due to the complex life cycle of the parasite and the absence of an effective vaccine.

### 1.1. Plasmodium Falciparum Life Cycle

*Plasmodiumfalciparum* leads a complex life cycle in both the vector, female *Anopheles*’ mosquito vector, and the vertebrate human host. The unicellular protozoan parasites are transmitted to humans by an infected mosquito when it takes a blood meal and injects the sporozoites into the dermis, which find their way into the bloodstream. The sporozoites circulate in the blood and enter hepatocytes, their first site of asexual reproduction [3]. The infected hepatocytes eventually rupture, releasing the merozoites into the bloodstream. The merozoites infect the red blood cells (RBCs) to initiate the intra-erythrocytic developmental cycle where the parasites start off in the ring stage and mature into trophozoites, and lastly, schizonts which rupture to release merozoites that reinvade new RBCs. In the infected RBC (iRBC), a small proportion of the ring stage parasites may also undergo sexual development to form gametocytes. These are ingested by the mosquitoes when they take a blood meal, thereby continuing the transmission cycle by forming gametes which start the sporogonic cycle within the mosquito [4]. However, the erythrocytic cycle of the parasite life cycle is mostly associated with malaria symptoms; for example, the adhesion phenomenon is associated with disease severity in *P. falciparum* [5]. This is attributed to the release of parasitic and iRBC intracellular material when schizonts rupture. These exposed parasites and intracellular materials stimulate immune responses that contribute to malaria pathogenesis [6]. During their complex life cycle, the parasites survive under a stressful environment as it transitions from the poikilothermic vector at 25 °C to the homoeothermic host at 37 °C. This results in thermal stress that is further increased during fever episodes associated with temperature spikes to 41 °C [7]. In order to survive these changes, the parasite needs a robust protein quality control system, and the parasite upregulates the expression of some of the molecular chaperones to maintain its proteome [8].

### 1.2. Antimalarial Drug Resistance

The treatment of malaria currently depends on Artemisinin-based Combination Therapy (ACT). However, its efficacy is under threat due to the increased clearance times observed with ACTs which are associated with the development of resistance to artemisinin monotherapy which has been reported in the Great Mekong region and recently in Africa [1]. This drug resistance trend is similar to the fate of chloroquine, where drug resistance started in the Mekong region in the early 2000′s prior to its widespread dissemination to various regions, including Africa and East Asia. Recently, artemisinin drug resistance parasite genotypes were identified in Rwanda, and this possibly suggests the beginning of widespread resistance development to artemisinin compounds [9].

The development of drug resistance is known to be due to parasite targeted gene mutations [10,11]. The proper functioning of mutated genes related to the influx and efflux pumps affect antiparasitic concentrations of the drug; for example, the correct folding of mutated *chloroquine resistance transporter (crt)* is attributed to molecular chaperones such as the Heat shock proteins (Hsp). These Hsps facilitate the native folding of virtually all proteins under stress, including drug pressure. In addition, the genes implicated in multidrug resistance, such as the *crt* gene, are located on the same gene cluster as Hsp90, due to the central role they play in protein quality control [12]. This suggests that the regulation through cis-regulatory elements such as transcription factors may similarly co-regulate the expression of both chaperone and the mutated *crt* gene. In addition, some molecular chaperones have been shown to, directly and indirectly, interact with crt through a yet to be established mechanism [10]. It is tempting to speculate that the chaperoning role of Hsps on proteins may promote drug resistance during the stress response. 

## 2. Heat Shock Protein 90 and Its Functional Cycle

Heat shock protein 90 (Hsp90) forms part of an evolutionary conserved and ubiquitously expressed group of molecular chaperones that make up about 2% of the cellular proteome [13]. Hsp90 proteins are found in almost all organisms, and they are essential for survival in eukaryotes but dispensable for survival under normal conditions in some eubacterial species [14,15]. The Hsp90 chaperone acts to reduce stress and maintain cellular homeostasis [16]. Hsp90 interacts with a variety of client proteins and serves as a central modulator of crucial cellular processes ranging from stress regulation, protein folding, DNA repair, development, immune response, and signaling pathways [14]. Due to the central role they play in cell survival, Hsp90s are attractive drug targets for several diseases, including cancer, neurodegenerative disorders, and infectious diseases, such as malaria [14]. 

In eukaryotes, Hsp90 family members are highly conserved across species and are expressed in different cell compartments like the cytosol (Hsp90), endoplasmic reticulum (glucose-regulated protein 94 (Grp94)), and mitochondria (Tumor necrosis factor receptor-associated protein 1 (Trap1)). Generally, Hsp90 exists as a V-shaped homodimer, and each protomer is composed of three structural domains, the N-terminal domain (NTD), the middle domain (MD), and the C-terminal domain (CTD; Figure 1A). The NTD serves to bind and hydrolyze ATP [16]. Hsp90 NTD shares structural features with the GHKL (gyrase, Hsp90, histidine kinase, MutL) family of proteins, which all have a “Bergerat fold” motif in the NTD that binds ATP [17,18]. The MD acts as the major binding site for client proteins and possesses a catalytic arginine group that functions to facilitate ATP hydrolysis [16]. A charged linker region connecting the NTD and MD together has been shown to modulate Hsp90 activity but is absent in mitochondrial and bacterial Hsp90 [19]. The C-terminal domain (CTD) comprises a major dimerization interface that allows Hsp90 to function as a fundamental homodimer [20]. Cytosolic Hsp90s terminate on a characteristic C-terminal MEEVD motif (Figure 1), which is the binding site for numerous co-chaperones that contain a tetratricopeptide repeat domain (TPR); a common structural motif that consists of two anti-parallel α-helices and is ~34 amino acids long [21]. On the other hand, ER-resident Grp94s have a C-terminal SDEL/KDEL ER-retention sequence that dictates their cellular localization [22]. 

Members of the Hsp90 family are ATP-dependent chaperones that function in several cellular processes, including cell cycle control, cell survival, hormone, and other signal transduction pathways [14,23]. The Hsp90 functional cycle is driven by ATP binding and the consequent hydrolysis thereof (Figure 2). During this cycle, in the ATP bound state, Hsp90 binds to an unfolded substrate/client protein in its open-dimer conformation [20]. This results in a conformational change that causes the N1, N4, and N5 helixes in the NTD to close over the ATP binding pocket (ATP-bound intermediate state), functioning as a lid over the cavity [24,25]. Thereafter, the NTDs associate and dimerize, which causes further conformational changes, resulting in the association of the MDs. These changes reposition the catalytic loop in the MDs. The catalytic loop spans residues 414 to 427 in *P. falciparum* and includes the critical catalytic residues Asn416, Arg419, and Gln423 that hydrolyze ATP [26]. Subsequently, ATP hydrolysis causes the dissociation of the NTDs opening the cavity, which allows for the release of the correctly folded client protein [27,28,29].

## 3. Plasmodial Hsp90s

The *P. falciparum* genome encodes for four *Hsp90* genes. The Hsp90 proteins are localized in different cell compartments, i.e., in the cytosol (PfHsp90; PlasmoDB accession number: PF3D7_0708400), the mitochondria (PfTrap1/PfHsp90_M; PlasmoDB: PF3D7_1118200), Endoplasmic reticulum (PfGrp94; PlasmoDB: PF3D7_1222300), and the apicoplast (PfHsp90_A; PlasmoDB: PF3D7_1443900) (Figure 1). The structure of cytosolic PfHsp90 is similar to that of canonical Hsp90, which terminates on an EEVD motif [26,29]. The ER-resident PfGrp94 has a split MD and terminates on an SDEL motif, which enables retention of the protein in the ER compartment (Figure 2). PfGrp94, as well as the apicoplast, localized PfHsp90_A, and the mitochondrial localized PfHsp90_M, all have N-terminal signal peptides that direct the proteins to their respective cellular compartments (Appendix A).

### 3.1. Cytosolic P. falciparum Hsp90

The cytosolic resident PfHsp90 share higher sequence identities with other cytosolic orthologues. For example, there is a 64% amino acid sequence identity shared between cytosolic PfHsp90 and cytosolic human HSPC2 (Figure 2). As such, the structural organization of PfHsp90 is closer to other cytosolic Hsp90 proteins from different organisms than its parasite paralogs located in other cell compartments [26,32]. Similarly, the PfHsp90 structure is composed of 3 main domains as canonical Hsp90, thus an NTD, MD, and CTD [33]. The PfHsp90 NTD region was shown to be the most conserved domain, sharing 75% identity with orthologous Hsp90 proteins, 56% identity with PfGrp94, 49% identity with PfHsp90_M, and 39% identity with PfHsp90_A as illustrated in Figure 2.

Among the *P. falciparum* Hsp90s, most of the attention has been to the cytosolic PfHsp90 (PF3D7_0708400), which has publicly available published crystal structures [26,34]. The crystal structure of PfHsp90 revealed that the N-terminal ATP binding domain consists of nine antiparallel β-sheets compacted with seven α-helices and forms part of the GHKL ATPase fold family [34]. However, PfHsp90 ATP binding was achieved without Mg^2+^ ions that coordinates β- and γ-phosphate oxygens to the oxygen side chain of Ans37 required for ATPase activity of the human Hsp90 [26,34]. Notably, this unique bonding causes the Asn37 side chain and the α- and β-phosphates of ADP to shift toward each other, possibly to facilitate ATP hydrolysis [35,36]. This suggests that the cytosolic PfHsp90 lacking a metal cofactor binds nucleotides in a unique conformation from other human Hsp90 proteins.

The PfHsp90 ADP bound monomer forms a unique structural configuration that is characterized by an ATP lid which is shifted away from the active site to facilitate ADP binding. This forms a distinctive straight conformation which opens up a new hydrophobic cavity as compared to the curved GHL ATP lid [37]. This PfHsp90-specific pocket of the GHL motif is delineated by the residues Gly100, Thr101, and Phe104 [37]. The position of the ATP lid is then restricted by crystal packing, thereby causing amino acids 118–121 to overlap with the ADP binding site, which in turn prevents nucleotide binding [34]. Interestingly, only in the structures of the *Saccharomyces cerevisiae* Hsp90s, Hsp82 and Hsc82, have an exclusive ATP binding pose where the ATP lid position tightly forms an envelope around the γ-phosphate of ATP to accommodate a hydrolytic conformation [34].

The charged linker, which connects the NTD and MD together, is the least conserved region amongst Hsp90 proteins [33]. PfHsp90 has a longer and more negatively charged linker region in comparison to the human HSPC2 and yeast Hsp82 [33,37]. This extended linker modulates several functions of the protein, such as domain flexibility, chaperone activity, client and co-chaperone binding, and consequently, the rate of ATP hydrolysis [38]. Interestingly, it has been observed that replacing the charged linker in yeast Hsp90 with the charged linker of PfHsp90 decreases client protein binding and ATPase activity [33]. This suggests that the charged linker plays a species-specific role in Hsp90 function and offers the PfHsp90 chaperone with unique features.

The MD of PfHsp90 has not been structurally characterized fully, but some functional similarities can be drawn from its orthologues. The Hsp90 is a secondary chaperone that facilitates folding and activation of more than 200 clients through the MD as a client binding site of the chaperone [20,39]. The communication between the NTD and MD is essential for function [19]. The MD dictates the overall chaperone function, including ATP hydrolysis, conformational changes, client, and co-chaperone binding [40]. The NTD modulates the structural dynamics of the client binding region in the MD. The MD, in turn, must physically interact with the NTD to facilitate ATP hydrolysis [40,41]. Therefore, this region acts as the sensor for the conformational signal encoded by nucleotides [42] and confers paralog specificity to Hsp90 [40,43].

The CTD of Hsp90 is the main dimerization domain [16]. It also possesses the conserved C-terminal pentapeptide MEEVD motif that facilitates binding to co-chaperones. The MEEVD motif interacts with TPR-containing co-chaperones that modulate the chaperone function and client fate [44,45].

### 3.2. Co-Chaperones of Cytosolic PfHsp90

The function of Hsp90 is regulated by its interaction with multiple co-chaperones to assist in client protein folding. Since Hsp90 conformation is dependent on its nucleotide-binding state, regulating its ATPase activity enables co-chaperones to facilitate structural changes that allow interaction and subsequent client activation [44,46]. It is, therefore, the coordinated binding of these co-chaperones that increase the efficiency and capacity of Hsp90s on client protein folding [47,48]. Different co-chaperones facilitate unique functions of Hsp90, thereby influencing its selection of clients. The Hsp90 protein associates with a variety of cytosolic co-chaperones illustrated in Table 1 [49].

The co-chaperone gene names with PlasmoDB accession numbers and functions are shown with the abbreviated protein names are as follows: FKBP35- immunophilin FK506-binding protein 35, PfCBP-calcyclin-binding protein, Cns1-cyclophilin seven suppressor 1; Hop/STI1 - stress-inducible protein homolog.

In *P. falciparum*, currently, only seven co-chaperones have been functionally confirmed to associate with PfHsp90. These may be grouped based on their binding domains to Hsp90. For example, most of the CTD binding co-chaperones possess TPR motifs that bind to the MEEVD motif of Hsp90, which include PfHop, PfPP5, and PfFKBP35 [29,48]. PfHop is an adaptor protein comprising of three TPR domains that bind to the C-terminal EEVD motif of PfHsp70 and MEEVD motif of PfHsp90 [44,50]. When bound, PfHop modulates the ATPase activity of the two chaperones to mediate the transfer of client proteins from PfHsp70 to the PfHsp90 machinery for protein folding and activation [44]. Two small acidic co-chaperones, Pfp23 orthologues, were identified in the *P. falciparum* genome [57]. Pf23 was shown to stabilize the dimerized form of PfHp90 in the presence of ATP [47]. Upon binding, Pfp23 suppresses the ATPase activity of PfHsp90 by trapping the chaperone in an ATP-bound state to prolong client protein binding and promote functional activation [47]. Similar to the Pfp23-PfHsp90 interaction, the two PfAha1 co-chaperones are also thought to bind PfHsp90 in an ATP-dependent manner. However, unlike Pfp23, PfAha1 binds to the MD of PfHsp90 and stimulates ATP hydrolysis [47]. In addition, it was found that Pfp23 and PfAha1 have overlapping binding sites as it appears as though the two co-chaperones interact mutually exclusively with PfHsp90 to regulate the chaperone function [47]. The PfPP5 is a protein metallo-phosphatase that is characterized by an N-terminal TPR domain and a C-terminal domain consisting of a Ser/Thr phosphatase catalytic core [64]. The TPR domain of PfPP5 is essential for binding to the MEEVD motif of PfHsp90 for potential de-phosphorylation of the chaperone [16,47]. PfHsp90 is phosphorylated to modulate its function [65]. This suggests that the co-chaperones phosphatase activity on Hsp90 plays an essential regulatory role in protein folding and maturation of Hsp90 client proteins [66,67]. The PfFKBP35 possesses a C-terminal TPR domain known to associate with PfHsp90 [68]. This co-chaperone also consists of an N-terminal domain that exhibits peptidyl-prolyl isomerase (PPI) activity which catalyzes protein folding and plays a role in the function of the immune system [68,69]. Genomic analysis has identified several putative PfHsp90 co-chaperones through protein domain homology, namely, PfCBP, PfPih1, and PfCns1 [47]. However, it still remains to be confirmed if these co-chaperones directly interact with PfHsp90 and their binding effects on the chaperone’s activity is unknown.

### 3.3. Plasmodium falciparum Endoplasmic Reticulum Hsp90 (PfGrp94)

The *P. falciparum* ER-localized Hsp90 (PfGrp94) is structurally composed of a similar domain architecture as the cytosolic orthologue PfHsp90. It has an additional 29 amino acids making up the N-terminal signal peptide followed by a pre-N terminal motif that is 40 residues longer than those found in the cytoplasmic paralogs (Figure 2). The pre-N-terminal motif contains many charged residues that are sensitive to proteolytic degradation and they regulate the rate of ATP hydrolysis [70]. The main catalytic residues involved in γ phosphate-binding are conserved on both the NTD and MD of PfGrp94 (Figure 3). The other notable variation from the cytosolic Hsp90 paralog is on the negatively charged linker connecting the MD to the NTD [31]. This linker sequence also varies greatly in length between species [19]. In PfGrp94, this region is shorter than in other Hsp90 paralogs, and it contains a single calcium-binding site. Calcium-binding is apparently required for peptide-binding by Grp94 [71]. The Grp94 MD is subdivided into two domains [72]. In other models, the MD is the main site for substrate binding, and it interacts with co-chaperones along with the NTD. Residues responsible for substrate interactions are well conserved between the orthologues except for Lys161, which is exchanged for Asn in PfGrp94 (Figure 3). The CTD dimerization domain has a signature interface comprised of four α-helix bundles, one from the MD and three from the CTD [73]. PfGrp94 contains an α-helix between residues 755 and 765 that extends the dimer interface and reinforces the stability of the dimer [74]. Furthermore, α-helix C2 does not form part of the dimerization bundle and instead extends out from this interface into the empty space between the two protomers [70].

The *P. falciparum* parasites remodel the iRBC by exporting over 400 proteins for their survival and reproduction [75]. The ER is the origin of the secretory pathway, where proteins are sorted and transported to various intra- and extracellular locations [76]. Most of the proteins destined for secretion and membrane-bound proteins carry N-terminal signal peptides that direct their co-translational import into the ER lumen, from where they are exported [77,78]. In the ER lumen, proteins attain their native tertiary structures through assisted folding by ER-resident molecular chaperones [79]. PfGrp94 is essential in parasite survival through its chaperoning role of client proteins destined for export and secretion [80,81]. This is consistent with the crucial role of Grp94 from other organisms such as mammals and *Drosophila* in the early developmental stages [82,83]. There is a need for further elucidation of PfGrp94 structure-functional features to characterize its interaction with accessory protein, including co-chaperones with orthologues in the cytosol.

### 3.4. P. falciparum Apicoplast Hsp90

The *P. falciparum* apicoplast-resident Hsp90, (PfHsp90_A), has not received much attention, and little is known about its function. However, functional insights for PfHsp90_A are drawn from the ER-resident Hsp90 paralog, PfGrp94, as both proteins are involved in protein folding in oxidizing organelles. It is assumed that PfHsp90_A is also involved in the folding of proteins under normal conditions and during times of stress in the apicoplast [84,85]. In other plastids, Hsp90s are needed to maintain proteostasis [86,87]. Unassembled, partially denatured, and misfolded proteins are either correctly folded by Hsps or they are marked for degradation by proteolysis [85,88]. Therefore, PfHsp90_A may also have additional functions, such as oligomer assembly and the targeting of proteins for degradation by the ER-associated protein degradation (ERAD) pathway in the cytosol, much like its ER-resident paralog [89]. There are no specific substrates identified for PfHsp90_A to date; hence, its function remains to be fully elucidated.

Structurally, the domain organization of the apicoplast -resident Hsp90 is the same as that of the other Hsp90 paralogs (Figure 2). It is unique to the other Hsp90 paralogs in that it has a bipartite NTD signal sequence to direct its transport over four membranes into the apicoplast. On the extreme C-terminal, the protein has a unique KTLL sequence, which may serve as an apicoplast retention signal, like the KDEL sequence of ER-resident proteins [90]. The preliminary crystal structure of PfHsp90_A N-terminal domain in complex with AMP phosphoramide (AMPPNP) has been solved and deposited on the protein data bank (PDB ID: 3IED, [91]. The structure has a unique ATP binding pocket in comparison to other Hsp90 paralogs. In addition, it was observed that the PfHsp90_A protein was co-crystallized with AMPPNP, in a similar way with a hydrolyzed form of ATP and AMP, without the γ-phosphate. This suggests that PfHsp90-A binds to AMPPNP and AMP in a similar fashion. The AMPPNP ligand, made contacts with the residues: Asn133, Asp211, Asn224, Phe257, and Thr308 that are conserved in all the Hsp90 paralogs and homologs (Figure 2). Three of these residues are part of the ATP binding pocket, suggesting that they are important for the binding and hydrolysis of ATP in the PfHsp90_A protein. The residue, Lys458 is involved in substrate binding in other orthologues but it is exchanged for a Ser residue in PfHsp90_A (Figure 3). The crystal structure of the PfHsp90_A MD was solved during genome-wide crystallization (PDB ID: 1Y6Z; [92]). An annotated structure is required to fully elucidate structural and functional features of the domain as the catalytic residues are conserved in other paralogs (Figure 3). Due to the high degree of structural similarities shared with other Hsp90 proteins, PfHsp90_A is expected to form functional dimers. 

### 3.5. P. falciparum Mitochondrial Hsp90 

The *P. falciparum* mitochondrial resident Hsp90, also known as Tumor necrosis factor receptor-associated protein-1 (PfTrap1), or PfHsp90_M, plays an important role in maintaining mitochondrial integrity and antioxidant activity [93,94,95]. PfHsp90_M functions are critical in the mitochondria as it chaperones the membrane transition pore proteins and electron transport chain protein components [96]. The efforts to fully characterize PfHsp90_M have been sluggish, and its functional and structural relationships remain unclear. PfHsp90_M localization in the mitochondria is facilitated by the presence of a unique N-terminal mitochondrial import sequence which is cleaved during organelle importation as in other homologs [97]. The high sequence similarity shared between PfHsp90_M with cytosolic PfHsp90 leads to the assumption that the structure organization between the two is somewhat shared (Figure 2). PfHsp90_M is composed of the same three structural domains as Hsp90 (Figure 3). Conversely, there are several significant functional differences between PfHsp90_M and other Hsp90 proteins. PfHsp90_M lacks both the C-terminal MEEVD motif and the charger linker segment [98]. The absence of the EEVD motif suggests that PfHsp90_M does not directly interact with TPR-containing co-chaperones for its functional cycle [99,100]. Potential co-chaperones for the mitochondrial Hsp90s still need to be identified both in the parasite and for the human HSPC5 homolog to better understand the protein function.

The mitochondrial Hsp90s have superior activity as they function in an oxidizing organelle. For example, the ATP hydrolysis rate of mitochondrial Hsp90 makes these proteins exhibit differences in catalytic efficiencies. The residues implicated in MD catalytic modulation are not conserved in PfHsp90_M as with other parasite paralogs (Figure 3). This suggests that, indeed, PfHsp90_M potentially exhibits unique catalytic activities. Previous studies using a human HSPC5 reported that HSPC5 exhibited a 10-fold higher binding affinity for ATP in comparison to the cytosolic human or yeast Hsp90 [96]. The HSPC5 NTD possesses divalent cation binding sites for magnesium and calcium [101]. The Mg^2+^ or Ca^2+^ cofactor binding stabilizes the closed N-terminal conformation of HSPC5, which is necessary for ATP hydrolysis [101]. Interestingly, the ATPase activity of HSPC5 is directly dependent on the concentration of calcium ions but not on magnesium ions. The effect of cations in the mitochondrial matrix on ATP hydrolysis rates of PfHsp90_M still needs to be confirmed as cation dependency potentially makes this protein an attractive drug target.

## 4. *P. falciparum* Hsp90 as Drug Targets

Most of the early inhibitors targeting Hsp90 function were towards cancer therapy. These inhibitors were mostly targeting the ATP binding pocket of Hsp90 NTD [102,103] due to cancer cells’ unquenchable addiction to Hsp90 [103,104]. Consequently, overexpression of Hsp90 in malignant cells points to its involvement in the maturation of numerous oncogenic client proteins. It is these features that make Hsp90 a desirable anti-cancer drug target [105]. Similar to other eukaryotes in *P. falciparum,* it is plausible that all four Hsp90s are essential for parasite survival and erythrocytic stage transitions [65,106]. However, it should be noted that most studies on parasite Hsp90s use known human Hsp90 inhibitors to determine the function and essentiality of parasite Hsp90s. There is a need for caution in interpreting such results as it has not been thoroughly validated that the effect of such inhibitors on parasite development is specific to Hsp90 and not due to other targets. Despite this limitation, there are several factors that make Hsp90 proteins attractive drug targets [107]. Firstly, Hsp90s are ATPases that are known to have varying degrees of activity in different organisms [108], and the rate of ATP hydrolysis is increased in diseased cells, making them more susceptible to ATPase inhibitors [109]. Similarly, *P. falciparum* Hsp90′s higher ATPase activity predisposes it to be more susceptible to inhibition than human homolog [10,106]. Secondly, Hsp90s associate with different co-chaperones in different organisms. These species-specific distinctive co-chaperone interactions could also be exploited for selective inhibition [29,110]. Lastly, it has been shown that the small differences in amino acid sequences resulted in some structural variations between Hsp90 proteins from different organisms and in different cellular compartments [111]. The elucidation of several Hsp90 crystal structures has provided more avenues to identify drug targets based on unique conformations of the Hsp90 proteins [112]. These structural insights gained from the high-resolution crystal structures have led to the screening and identification of novel Hsp90 inhibitors. Several crystal structures of different Hsp90 NTDs in complex with nucleotides [34,46,113] and inhibitors [114,115] have been solved. In addition, full structure characterization of Hsp90 has led to the design of inhibitors targeting the MD [26,116,117,118] and CTD of Hsp90 [119,120]. Some of these inhibitors have been shown to be effective at inhibiting *P. falciparum* parasite growth through targeting Hsp90 function [10,31,121,122,123].

The most studied small molecule inhibitors targeting the Hsp90 NTD are ATP mimetics. These mimetics comprise several groups of molecules that structurally resemble nucleotides [24,31]. Inhibitions occur when Hsp90s are inactivated by these compounds through competitively binding to the ATP binding pocket in the NTDs when the proteins are in the open conformation [10,106,124,125]. This traps the Hsp90 protein in the inhibitor-bound state and renders it unable to complete its ATPase cycle [126]. As such, the most common Hsp90 enzymatic activity assay employed is the ATPase assay in parasite Hsp90 inhibition studies [8,10,106]. This, by extension, results in Hsp90’s failure to facilitate the folding of substrates to achieve their native conformations in the presence of inhibitor [127]. Consequently, there is an ensuing accumulation of unfolded proteins in the cell [128]. Such unfolded proteins may aggregate, causing toxicity and, inevitably, cell death [129]. A variety of ATP mimetics have been developed and are currently the most effective drugs being tested at different stages in preclinical trials in cancer with the potential for repurposing to malaria [10,111,130,131]. These include natural inhibitors such as ansamycin, radicicol, harmine, acrisorcin, and synthetic inhibitors such as purines, pyrazoles isoxazoles, and other scaffolds.

### 4.1. Inhibitors of the Hsp90 N-Terminal Domain

#### 4.1.1. Geldanamycin and Its Derivatives

Geldanamycin (GA), a benzoquinone ansamycin compound naturally produced by *Streptomyces hygroscopicus* and it was among the first Hsp90 inhibitors to be identified (Figure 4) [102]. GA was originally thought to be a kinase inhibitor antibiotic, but it was later established that it specifically binds to Hsp90 with high specificity [125,126]. Most of these early studies focused on targeting Hsp90 in cancer cells and later repurposed to other diseases, including malaria. As such, some of the repurposed cancer therapeutics and inhibitors were shown to exhibit potent antiplasmodial activity [65,121]. In *P. falciparum*, GA competitively binds to the ATPase domain of PfHsp90 [132]. Subsequently, the presence of GA abrogates the association of PfHsp90-client protein and results in degradation of the unfolded client protein [65,130]. GA inhibited in vitro parasite growth at an IC_50_ comparable to that of the known antimalarial chloroquine (20 nM and 15 nM, respectively) [65,130]. This inhibition resulted in a stage transition arrest for intra-erythrocytic parasite stages, mainly affecting the transition from ring stage to trophozoite stage, which suggests that PfHsp90 plays an essential role in parasitic development [65,133]. Additionally, GA has also been reported to be equally effective against both chloroquine-resistant and chloroquine-sensitive strains [133]. The binding affinity of GA for cytosolic PfHsp90 was much higher, at a lower micromolar range, when compared to its paralog PfGrp94 and human ortholog HSPC2, which were in the millimolar range [81]. Similarly, an independent study observed GA to be more effective at inhibiting the ATPase activity of PfHsp90 than human HSPC2 [106]. This suggests that GA is more selective to PfHsp90 enzymatic function abrogation than the human equivalent. However, GA lacked binding selectivity between the parasite ER paralog PfGrp94 and the human cytosolic HSPC2 as both had affinities for GA in the millimolar range [81]. This suggests that GA inhibits human cytosolic HSPC2 and PfGrp94 similarly, which could result in high toxicity and inadvertent side effects if PfGrp94 were to be targeted. 

Despite the promising indices from GA, its poor solubility and high hepatotoxicity were the major limitations for approval in clinical testing as both an anticancer and antimalarial drug [98,134]. These limitations, in part, led to the development of several GA derivatives with improved drug-like indices, such as low toxicity profiles and increased in vivo activity and stability. The GA derivative, 17-allylamino-17-demethoxy geldanamycin (17-AAG, KOS-953; CNF; tanespimycin), which has a modification on the carbon 17 position of the Ansa ring (Figure 4, [135]), exhibited increased potency and less toxicity than GA. This compound, 17-AAG, showed promising results against various cancers in several phase I trials, reviewed in [136] and a phase II clinical trial against HER-2 breast cancer [137]. The repurposing of 17-AAG towards malaria was not very effective as it lacked selectivity for parasite Hsp90s. Thus, 17-AAG has a higher affinity for human HSPC2 (0.09 µM) than with parasite Hsp90 (PfHsp90 at 4.54 µM and PfGrp94 at 28.5 µM) [10,81]. In addition, 17-AAG had poor IC_50_ values in parasite cultures compared to GA. Again, due to poor solubilities 17-AAG, was further modified to yield 17-dimethylamino ethylamino-17-demethoxygeldanamycin (17-DMAG, KOS1022; alvespimycin), which contains an ionizable group on the carbon 17 with improved water solubility. The binding affinity of 17-DMAG to parasite Hsp90 (PfHsp90 and PfGrp94) was at least one order of magnitude stronger than for the human HSPC2 homolog (8 nM and 23 nM against 158 nM, respectively) [81]. These findings were in contrast with reports from another study which used the full-length recombinant PfHsp90 protein that did not observe significant differences between the 17-DMAG binding affinities for PfHsp90 and human HSPC2 [121]. These conflicting results may be due to the use of recombinant full-length protein in one study [121] and the truncated NTD protein version in the other study [81]. This suggests that the NTD in isolation exhibits unique binding affinities that are not observed in the full-length version. Thus, as the Hsp90 MD is responsible for activating ATPase and acts as a sensor for nucleotide-binding on the protein, its removal affected the function of the NTD in isolation [42]. This further suggests that the in vivo subdomain data should be interpreted with caution despite the valuable information that can be deciphered from these analyses. 

*P. falciparum* parasite growth inhibition studies with GA and its two derivatives 17-AAG and 17-DMAG, were not as effective as chloroquine on both the *P. falciparum* 3D7 chloroquine-sensitive strain and the chloroquine-resistant strain *P. falciparum* W2 [81]. Hence, further studies on the 17-AAG modified version, 17-allylamino-17-demethoxygeldanamycin hydroquinone hydrochloride (IPI-504, retaspimycin hydrochloride) [138] and 17-amino-17-demethoxygeldanamycin (17-AG, IPI-493), which is a metabolite of IPI-504 and 17-AAG [117], should be prioritized. Both IPI-504 and IPI-493 have promising prospects as Hsp90 inhibitors in cancer [117,138], and they provide a scaffolding platform for derivatization towards parasite selective Hsp90 inhibitors.

#### 4.1.2. Harmine and Its Derivatives

The rapid development of in-silico drug design was made possible by the availability of a high-resolution crystal structure of Hsp90NTD in a complex with nucleotides and GA [139,140]. In order to generate more parasite selective synthetic inhibitors for Hsp90, high throughput screening (HTS) of inhibitors was employed [10,141]. Promising parasite Hsp90 selective inhibitors were developed, including acrisorcin, harmine, and 2-amino-3-phospho propionic acid (APPA) with an IC_50_ of 51.3 nM, 50.3 nM, and 60.3 nM, respectively (Figure 5) [10,141]. The selective binding of these inhibitors to parasite Hsp90 over human Hsp90 was due to their direct interaction with PfHsp90 Arg98 over the human Hsp90 Lys90 at the same position [141]. These findings made it attractive to modify these inhibitors to make them more selective and improve the drug indices.

Derivatization on the harmine scaffold generated harmanol, which had improved selectivity with a higher binding affinity to parasite Hsp90 than for the human Hsp90. Interestingly, these inhibitors had a synergistic effect with chloroquine to suppress chloroquine resistance development [10]. Further derivatization of harmine by combining the β-carboline alkaloid harmine through a triazole linker to cinnamyl azides yielded three potent PfHsp90 inhibitors, namely: N-harmicines 6, O-harmincines 7, and *N,O*-*bis*-harmicines 8 (Figure 5) [142]. The antiplasmodial activities of these compounds were higher than the parent harmine towards chloroquine-sensitive *P. falciparum* 3D7 and chloroquine resistance *P. falciparum* Dd2 parasite strains. Using molecular dynamics simulations, the residues implicated in improving binding affinities for *N,O-bis*-harmicines were the methoxy group in 8b, which was stabilized by Asp79 [142]. As such, to utilize the triazole ring that was not contributing to PfHsp90 binding, a triazole was replaced by an amide bond, its bioisostere, to produce amide-type harmicides 5 and 6 [143]. These *N*-harmicines 5 and *O-*harmicides 6 compounds had higher antiplasmodial activity than the triazole-type harmicines. The most active compound against both chloroquine-sensitive and resistant strains, *P. falciparum* 3D7 (IC_50_ 040 nM) and Dd2 (IC_50_ 170 nM), respectively, was *N-*harmicine 5e [143]. Notably, both compounds 5b, and 5a showed the highest selectivity indices (ratio of IC_50_ of the parasite cell line *P. falciparum* 3D7 compared to host cell line HepG2) with SI >700. These were followed by compounds 5f and 5d (SI 287 and 233) and 5e with (SI = 73). The selectivity indices for the *N*-harmicines were higher than the *O*-harmicines derivatives. Molecular dynamics simulations confirmed the binding trends as 5e > 5d > 5a which were higher than the matching *O*-harmicines 6e > 6d > 6a. These simulations predicted compound 5e to be positioned inside the ATP binding site, which was different from the predicted harmine binding site, that is outside the ATP binding pocket [143]. Further modifications to these harmicines produced harmicine 27a m(triafluoromethyl)cinnamic acid, which interacted with ATP binding residues (Arg98, Asn37, and Phe124; [126]), and proved to be more selective against *P. falciparum* (SI=1105) [144]. This was due to minimizing steric and electronic hindrance from the ATP binding residue Asp79 [144]. Harmicine 27a was more selective against *P. falciparum* (SI= 1105; [144]). These studies revealed that amide-type harmicides modified at the N-9 of the β-carboline ring are the most potent with IC_50_ in the nanomolar range against the erythrocytic stages of the *P. falciparum* parasites and submicromolar range for the liver stage model *P. berghei* parasites [144]. However, the compounds had diminished potency on chloroquine-resistant parasite strain in the sub-micromolar range similar to chloroquine. This suggests that although promising, these inhibitors may have other properties that can be circumvented by the parsites, in the same way as chloroquine. It is tempting to speculate that drug resistance to Hsp90 inhibitor by chloroquine resistant parasites is modulated through the interaction between the crt and PfHsp90 [10]. Therefore, there is a need to further elucidate the mechanism of resistance and to modify these promising compounds as potential antimalarial drugs. 

In a recent study, virtual screening of compounds on the ZINC15 database [145] identified three potent novel compounds, CP-6, CP-7, and CP-10, that target the ATP binding pocket of PfHsp90 [146]. These three compounds selectively inhibited the growth of chloroquine-sensitive *P. falciparum* 3D7 strain without toxicity to human fibroblast BJ cell line [146]. This highlights the ability of these antimalarials to specifically target PfHsp90 without inducing cytotoxicity in the human host. Of these three hits, CP-7 was the most potent due to strong binding through the 1,3-benzodioxole ring of CP-7forming hydrogen bonds with Thr101 near the phosphate-binding site of the ATP pocket which is further strengthened by non-polar interactions with Tyr125 [146]. The central amide of CP-7 also forms an H-bond with the Asn37 primary amide, which makes stronger interactions resulting in higher potency against *P. falciparum* parasites [146]. This suggests that these three compounds provide a scaffolding platform for promising antimalarial drug candidates. 

#### 4.1.3. Purine Scaffolds

There have been several attempts to use purines as potential Hsp90 inhibitors. These purine scaffolds mainly target a unique ATP binding fold identified through crystal structure analysis of the Hsp90NTD complexed to ADP [140] and its inhibitors, GA [139] and Radicicol [103,126]. This distinct fold allows ATP to conform to a unique bend which was used for modifying the purine scaffold to yield PU3. PU3 possesses a purine core linked to the aryl moiety with methylene linker via carbon 8. This allows PU3 to mimic the bent shape of ATP bound to the Hsp90 NTD [147]. Based on these structure-based drug designs, PU3 had better affinity compared to GA due to its purine linker aryl motif. Further derivatization of purine scaffolds was mainly focused on changing the 8-aryl ring and the 9-alkyl substituents, which were amenable to increase affinity to bind Hsp90 (Figure 6; [148]). These modifications resulted in several drugs that proceeded to cancer clinical trials with improved pharmacokinetics and toxicity profiles, including PU-H71, MPC-3100, CNF2024/BIIB021, and debio0932 [148]. Of these, PU-H71, was more promising as it was synthesized to increase the binding to the Hsp90 NTD ATP binding cleft by using a 2-iodo-4,5-methylenedioxy for the aryl group (Figure 6; [121]. As with other anticancer Hsp90 inhibitors, the promising purines were evaluated towards repurposing as potential antiplasmodial Hsp90 inhibitors. The purine PU-H71 was shown to be effective as an antiplasmodial Hsp90 inhibitor in *P. berghei* malaria models and synergized with chloroquine to improve rodent survival [10]. As with other Hsp90 inhibitors, cross-reactivity with human Hsp90 limited the clinical prospects of this derivative as an antiplasmodial candidate. 

#### 4.1.4. Other In Silico Derived Scaffolds

In order to produce inhibitors that are more parasite selective, Wang and colleagues [122] used the PfHsp90NTD crystal structure (3K60; [34]) to target a unique hydrophobic pocket that extends from the base of the conserved Hsp90 ATP binding pocket [122]. This unique PfHsp90 hydrophobic pocket set it apart from the other Bergerat fold ATPases [34,122]. Virtual screening for potential small molecule inhibitors led to the identification of an effective 7-azandole derivative, methyl 1-isopentyl-3-(2-methoxyacetamido)-5-((6-methylhept-5-en-2-yl)amino)-1H-pyrrolo(2,3-b)pyridine-2-carboxylate (IND31119) and two amino-alcohol-carbazole (N-CBZ) compounds 5B and 5E with PfHsp90 selectivity (Figure 6; [122,123]). The three inhibitors, IND3119, compound 5B, and 5E had superior selective PfHsp90 binding affinity coupled to selective *P. falciparum* parasite growth inhibition when compared with GA [122,123]. The nanomolar range parasite growth inhibition suggests that compounds 5B and 5E possibly interfered with other cellular processes in the parasite in addition to those dependent on PfHsp90. There is a possibility that compounds 5E and 5B also inhibited other Hsp90 isotypes in different parasite cell compartments, which may share the unique hydrophobic pocket.

The Hsp90 paralogs can be selectively targeted using their different biological activities in response to stress. Efforts to design paralog selective parasite Hsp90 inhibitors are still in infancy, but there is promise from cancer studies. For example, HSPC5 is overexpressed in the mitochondria of tumor cells, whilst the other paralogs are low in normal tissues [149]. Therefore, to target the mitochondrial Hsp90 paralogs, another group of inhibitors such as Gamitrinibs, which are formed by GA C-17 connected to a mitochondrial targeting moiety, have gained attention (Figure 6 [150,151]). The Gamitrinibs show acute mitochondria toxicity through their inhibition of HSPC5. This leads to a loss of membrane potential and the release of cytochrome c, resulting in selective cancer cell apoptosis [150,152]. Another purine-scaffold Hsp90 paralog selective inhibitor of Grp94, N-ethyl-carboximido adenosine (NECA), was identified (Figure 7) [112,113]. The selectivity of NECA and its derivatives N-propylcarboxamido adenosine (NPCA), N-hydroxyethylcarboxamido adenosine (NEoCA), and N-aminoethycarboxamido adenosine (NEaCA) [112] for Grp94 are based on the N-ethyl carboximido moiety that extends into the third hydrophobic pocket that surrounds the ATP binding pocket of Grp94. Furthermore, three purine derivatives, 8-((2,4-Dimethylphenyl)thio)-3-(pent-4-yn-1-yl)-3H-purin-6-amine (PU-H54), 6-amino-8-(C3,5-dichlorophenyl)thio)-N-(1-methylethyl)-9H-purine-9-propanamine (PU-WS13) and 8-((2,4-Dichlorophenyl)thio)-9-(pent-4-yn-1-yl)-9H-purin-6-amine (PU-H39) were identified as potential paralog selective inhibitors (Figure 6) [99,149]. These compounds, PU-H54, PI-WS13, and PU-H39, had a 100-fold selectivity towards Grp94 over the cytosolic paralogs and 10–100-fold selectivity over the mitochondrial paralog (HSPC5) [149]. The selectivity was due to the compounds getting access through Phe199 to a binding pocket which is somehow blocked in the cytosolic paralog by Phe138 side chains [149]. It is tempting to speculate that mirroring such cell compartment targeting drugs to unique parasite organelles offers promise for potential drugs, considering that the parasite mitochondrion and ER play a significant role during survival under drug pressure. 

#### 4.1.5. Radicicol and Its Derivatives

Radicicol (RA) is a natural ATP mimicking antibiotic with a 14-carbon macrocyclic lactone structure. Radicicol has limited in vivo use as it is chemically and metabolically inactive in cells, despite its potent in vitro Hsp90 inhibitory properties (Figure 7; [153]). However, the elucidation of radicicol’s binding mode on human and fungal Hsp90 by X-ray crystallography and NMR [154,155], heightened the interest in the development of radicicol derivatives with improved activity and stability in cells [156,157]. The repurposing of radicicol as a potential *P. falciparum*, Hsp90 inhibitor was motivated by its capability to specifically inhibit schizont parasite stage development. This was thought to be through radicicol activity in abrogating parasite mitochondrial replication [158]. In addition, findings from direct binding assays suggested that radicicol elicited its antiparasitic activity through binding to either of two potential target genes, topoisomerase VIA /VIB [158,159] or to the mitochondrial PfHsp90_M (Trap1; [160]). The side-by-side comparative binding analysis between these genes has not been confirmed. However, this suggests that the potential radicicol cross-reactivity with both PfHsp90_M and topoisomerases is based on the shared Bergerat ATP binding pocket. With this promising evidence, there has been revived interest in radicicol scaffold modifications towards targeting topoisomerases [160]. It remains to be validated if these radicicol derivatives targeting isomerases do indeed cross-react with parasite Hsp90s.

Derivatization of radicicol to generate Hsp90 paralog selective inhibitors yielded radamine and resorcyclic pyrazoles/isoxazoles. Radamide and the other derivative compounds 7.1, 7.2, 7.4, 7.5, 7.6 and 7.7 were selective for human HSPC4 [161,162]. The HSPC4 crystal structure in complex with radamide and compound 7.7 has been solved [113,162,163]. The structure elucidation revealed that for selectivity, these RA derivatives form hydrogen bonds with the carboxylate group of Asn107, Asp149, Lys168, and Thr245 of HSPC4 [162,163]. These residues line the Grp94 ATP binding pocket and occupy a unique small hydrophobic pocket that is not present in other Hsp90 paralogs [164]. It remains to be validated if parasite Grp94 can be specifically targeted as the RA derivatives binding residues are conserved in the parasite PfGrp94 and PfHsp90_A paralogs. This suggests that there are prospects for both radamine and compound 7.7 towards targeting parasite Grp94, which may result in antiplasmodial activity. Preliminary studies identified NVP-AUY922, also known as luminespib (Figure 7), a resorcinylic isoxazole amine that binds and inhibits Hsp90 with a higher affinity than radamide [165]. The binding affinity for luminesbip to PfGrp94 was 10-fold higher compared to the cytosolic PfHsp90 and HSPC2 [81]. Despite having promising selectivity against PfGrp94, the similar high binding affinities with both PfHsp90 and HSPC2 limited its prospects in the current form. Nevertheless, this creates more opportunities for further derivatization to develop selective inhibitors against parasite Hsp90s. 

### 4.2. Middle Domain and C-Terminal Domain Inhibitors

Most of the prospective Hsp90 MD and CTD inhibitors available target the cytosolic Hsp90 paralog. Although the development of CTD inhibitors is in its early stages, it has become an emerging novel approach to inhibit Hsp90 chaperone function (Figure 8; [166]). The first discovered C-terminus inhibitor of Hsp90 was Novobiocin, an antibiotic from the coumermycin family, which is composed of a benzamide side chain, coumarin core, and a noviose sugar [167]. Novobiocin binds to the Hsp90 C-terminal region between residues 538–728, which harbor the second ATP binding site near the Hsp90 dimerization domain [167,168]. Due to the low-affinity binding of novobiocin to the Hsp90 CTD, attempts to improve the inhibitory activity of novobiocin on Hsp90 have been undertaken. The derivatization of novobiocin resulted in a promising potent compound A4, which is composed of a shorter N-acyl side chain in the benzamide structure and without both a 4-hydroxy substituent in the coumarin moiety and a carbamoyl group on the noviose sugar [169]. Further derivatization resulted in 4-deshydroxynovobiocin (DHN1) and 3′-descarbamoyl-4-deshydroxynovobiocin (DHN2), which exhibited higher potency (IC_50_ of 7.5 µM and 0.5 µM respectively) when compared to novobiocin [170]. Other C-terminal Hsp90 inhibitors that form part of the coumermycin family include clorobiocin, and coumermycin A1 derivative [171]. Other promising CTD inhibitors include 3,4-dihydropyrimidin-2(1H)-one (DHPM) [166,172]. 

Despite the numerous inhibitors that have been designed to target the NTD and CTD of Hsp90, very few studies have focused on the development of small-molecule inhibitors targeting the MD. A natural product, gambogic acid (GBA), selectively binds to a unique binding pocket in the MD of the human cytosolic HSPC3 paralog [173]. Docking studies revealed a potential MD binding pocket situated at residues 350–436 of HSPC3. A hydrophobic cavity at residues Ile353, Leu369, Ile370, and Ser365 is in the binding pocket, which can accommodate the hydrophobic side chain of GBA [173]. The GBA forms H-bonds with residues Glu372, Asn436, Asn375, and Asn351, which further stabilizes its association with HSPC3. Conversely, GBA and its derivative DAP-19, with a morpholine amide in replacement of the carboxylic acid were selective for the Hsp90 paralog and did not bind HSPC2, HSPC4, and HSPC5 [173]. Recently, Zhang and co-workers demonstrated that Triptolide (TL) is a Hsp90β inhibitor that targets the MD [116]. Triptolide, a diterpene triepozide, is a natural compound that exhibits antiproliferation, immune modulation, and anti-inflammatory activity [116,174]. These studies show the prospects of selective inhibition of Hsp90 paralogs as the MD is not highly conserved among the parasite Hsp90s and is the driver for paralog-specific functions. Hsp90 MD and CTD have not received much attention in small molecule inhibitor designs. This may soon change, as recently, the full-length high-resolution structure of PfHsp90 in solution was solved [26]. This will serve as a platform for accelerated structure-based Hsp90 MD and CTD inhibitors development through in silico docking studies.

### 4.3. Inhibitors Targeting Co-Chaperone Interactions

The majority of the inhibitors targeting co-chaperones of cytosolic Hsp90 are synthetic peptides that bind to the MEEVD motif at the C-terminus end [175]. These peptide inhibitors are TPR-based mimetics that compete for the Hsp90 MEEVD motif and prevent TPR co-chaperone interaction. The most common approach target the main TPR co-chaperone, Hop. This is achieved by utilizing TPR peptide analogs that can abrogate the association of Hop with Hsp90 chaperones [176,177]]. Based on the importance of residues Lys301 and Arg305 on the TPR2A domain of Hop, a peptidomimetic compound (Antp-TPR) inhibited Hsp90 binding [176]. The Antp-TPR peptide exhibited similar Hsp90 binding affinities when compared with full-length Hop. These high binding affinities resulted in promising anti-tumor activity across a variety of cancer cell lines through the degradation of cancer-related Hsp90 client proteins such as Akt and CDK4 [175]. As a result, the other TPR-based co-chaperones were targeted. For example, a small molecule compound, 1,6-dimethyl-1-3-propylpyrimido(,4-e)(1,2,4)triazine-5,7-dione (C9) was reported to reduce the levels of the Cdc37 [177]. The natural products, sansalvamide A (San-A) compounds, i.e., San-A 1 and San-A 2, abrogated the interaction of both FKBP38 and FKBP52 with Hsp90 at a higher efficacy in comparison to Hop [178]. This could be due to subtle differences in the interaction interfaces or binding sites of these TPR co-chaperones and Hsp90 [176]. In addition, other examples of inhibitors that target co-chaperone interaction with Hsp90 include the natural compound celastrol [179], cucurbitacin D [180], and gedunin [181], which were shown to disrupt the binding of co-chaperones Cdc37 and p23. Although Aha1 co-chaperone does not contain TPR repeats, it remains a promising drug target due to its role as the most potent activator of Hsp90 ATPase activity and binds to both the NTD and MD of Hsp90 [182]. Currently, there are no validated small molecule inhibitors of the PfAha1 co-chaperone, but in other systems, Hsp90-Aha1 ATPase activation inhibitors have been proposed [183]. Studies have shown that the inhibition of the c-AbI kinase prevents Aha1 phosphorylation which therefore blocks the co-chaperone from interacting with Hsp90 [184]. Taken together, the repurposing of inhibitors that target the parasite Hsp90-cochaperone interactions offers more opportunities for antimalarial drug design.

## 5. Conclusions and Future Perspectives

When geldanamycin was first identified as a potent Hsp90 inhibitor, its derivatization soon ensued, and many more Hsp90 inhibitors were developed. These efforts were mainly driven by the essential role that Hsp90 plays in eukaryotes and its implication in various diseases, such as numerous cancers and parasitic diseases [185,186]. The biggest limitation with Hsp90 inhibitors apart from drug-like properties has been that they target closely related paralogs due to the high degree of sequence and structural similarities between paralogs and orthologs. However, selective inhibitors have been developed to circumvent cross-reactivity, but their application in inhibiting the parasite Hsp90 has been limited. In clinical trials, the simultaneous inhibition of Hsp90s paralogs and off-targets in anticancer treatments were the main causes of hepatotoxicity and adverse side effects in patients. The story could be different in parasites; if the inhibitors can be made selective to target all parasite Hsp90 paralogs and not cross-react with the host Hsp90, then these drugs will be more attractive as antimalarials. This is plausible as a purine scaffold derivative, DN401, is a promising pan-Hsp90 inhibitor that abrogates the function of HSPC2, HSPC3, HSPC4, and HSPC5 paralogs [111]. Derivatization of DN401 to be selective for parasite Hsp90 offers more prospects in parasite inhibition. Recently, design and screening for potential parasite pan-Hsp90 inhibitors are gaining attention. A promising compound, violacein, was shown to target PfHsp90 and PfHsp70, thereby abrogating the chaperone function of the molecules with promising antiplasmodial indices [187]. Since the ATP binding of PfHsp90 is closer to that of its paralogs, as they share the same Bergerat fold than to Hsp70s [34], this suggests that violacein potentially inhibits other parasite Hsp90s in eliciting its anti-plasmodial activity. On the other hand, with high sequence similarities between Hsp90s localized in equivalent cell compartments from different species, the development of paralog-specific Hsp90 inhibitors is an attractive avenue worth exploring. As such, several Hsp90 paralog selective inhibitors have been developed [112,162,163,187,188]. These paralog selective inhibitors, if adopted to the parasitic PfHsp90_A apicoplast resident chaperone, may offer interesting possibilities, since this target is only present in apicomplexans and not in humans.

Peptidomimetics are potential drug candidates for inhibiting the direct association of Hsp90 and its co-chaperones and client protein interaction sites. The specificity of Hsp90 for clients is dictated in part by their interaction with co-chaperones. As such, peptide-based molecules that mimic co-chaperone interaction sites such as the MEEVD motif of Hsp90 and TPR domains represent attractive drug design opportunities. In addition, these protein-protein interaction inhibitors may be modified to be species-specific by manipulating the unique co-chaperone complements in different species. Furthermore, these peptide mimetic inhibitors may target the chaperone-client interaction sites to selectively inhibit Hsp90 function with minimal cross-reactivity with the host chaperone system. For example, the parasite proteome is known to be unique by possessing at least 24% asparagine-rich repeats [189,190]. These motifs can be targeted without affecting the host proteome. A potential obstacle with such peptide-based inhibitors is that they the challenge of bioavailability and membrane permeability must be addressed, as the parasite is localized intracellularly [191,192,193]. For drugs to effectively target malaria parasites, they first must cross three membranes: the host RBCs membrane, the parasitophorous vacuole membrane, and the parasite plasma membrane for cytosolic Hsp90. Additionally, a fourth organelle membrane must be crossed for apicoplast, ER or mitochondria for PfHsp90-A, PfGrp94 and PfHsp90_M, respectively [194]. This is a major setback to targeting organellar Hsp90s. However, drugs can be targeted to various intracellular compartments through cell trafficking principles: for instance, by modifying drugs to include appropriate transport signals [195] to target it to the desired intracellular compartment, without compromising the drug.

The modern drug pipeline for Hsp90 involves the use of high throughput screening of hundreds of thousands of compounds which can be conducted in silico [10,123,196]. These drug screening efforts depend mostly on using the high-resolution 3-dimensional structures of Hsp90 proteins in complex with nucleotides or natural and synthetic inhibitors. The limited availability of parasite Hsp90 paralogs with high-resolution 3-dimensional structures that are publicly available has curtailed progress in parasite paralog selective Hsp90 inhibitor development. It is interesting to note that the recent availability of high-quality 3-dimensional structure models of these paralogs solved by Alpha fold [197,198] may increase the pace for selective inhibitor design. This brings new dimensions to the highly sought-after 3-dimensional structures for in silico inhibitor design, which may potentially herald exponential growth in antiplasmodial drug design efforts. As with the current combinatorial treatment against malaria, it offers hope should novel inhibitors of parasite Hsp90 be identified. Such inhibitors may be used in combination with current treatments to minimize drug resistance development. 

## Figures and Tables

**Figure 1 cells-10-02849-f001:**
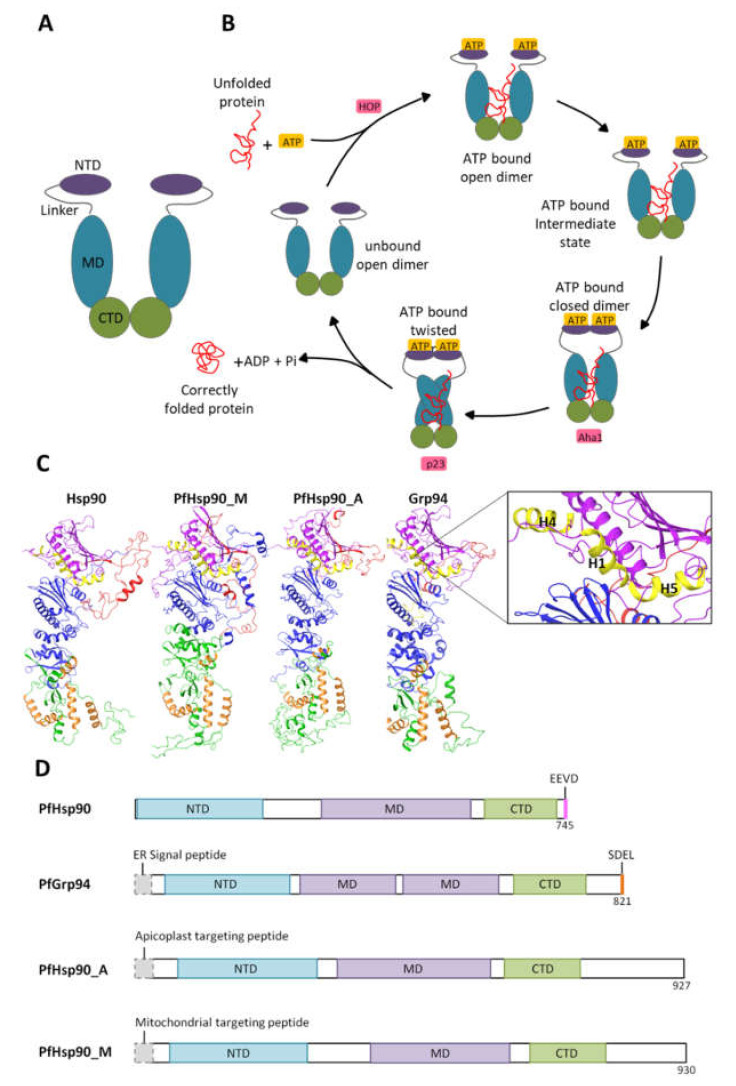
The nucleotide-dependent Hsp90 functional cycle and domain organization of *P. falciparum* Hsp90s. (**A**) Schematic of the general Hsp90 protein domain organization. (**B**) The Hsp90 functional cycle begins when Hsp90 is bound to ATP associate with an unfolded/partially folded client protein. Subsequently, the lid region closes over the ATP binding pocket, and the NTD dimerize, adopting a closed conformation. When the MDs associate, there is a repositioning of the catalytic loop in the MDs, which enables ATP hydrolysis. Upon ATP hydrolysis, the correctly folded client protein is released. The Hsp90 homodimer returns to the unbound open conformation and is primed for subsequent rounds of ATP hydrolysis and protein folding. This functional cycle is also modulated by co-chaperones such as HOP, Aha1, and p23. (**C**) Hsp90 protein models with the domains color codedthe NTDs are (purple), the linker (red), the MDs (blue), and the CTDs (green). The NTD helices H1, H4, and H5 are shown in yellow. Helices involved in dimerization are shown in orange. (**D**) The schematic of *P. falciparum* Hsp90 linear domain architecture for the differently localized Hsp90 proteins. Three-dimension (3-D) protein structures were generated by depositing protein sequences to the online PHYRE2 protein fold recognition server (http://www.sbg.bio.ic.ac.uk/phyre2/html/page.cgi?id=index, accessed on 18 April 2021). The 3-D model structures were visualized using the Schrodinger Maestro release 2021-3: LLC, New York, NY.

**Figure 2 cells-10-02849-f002:**
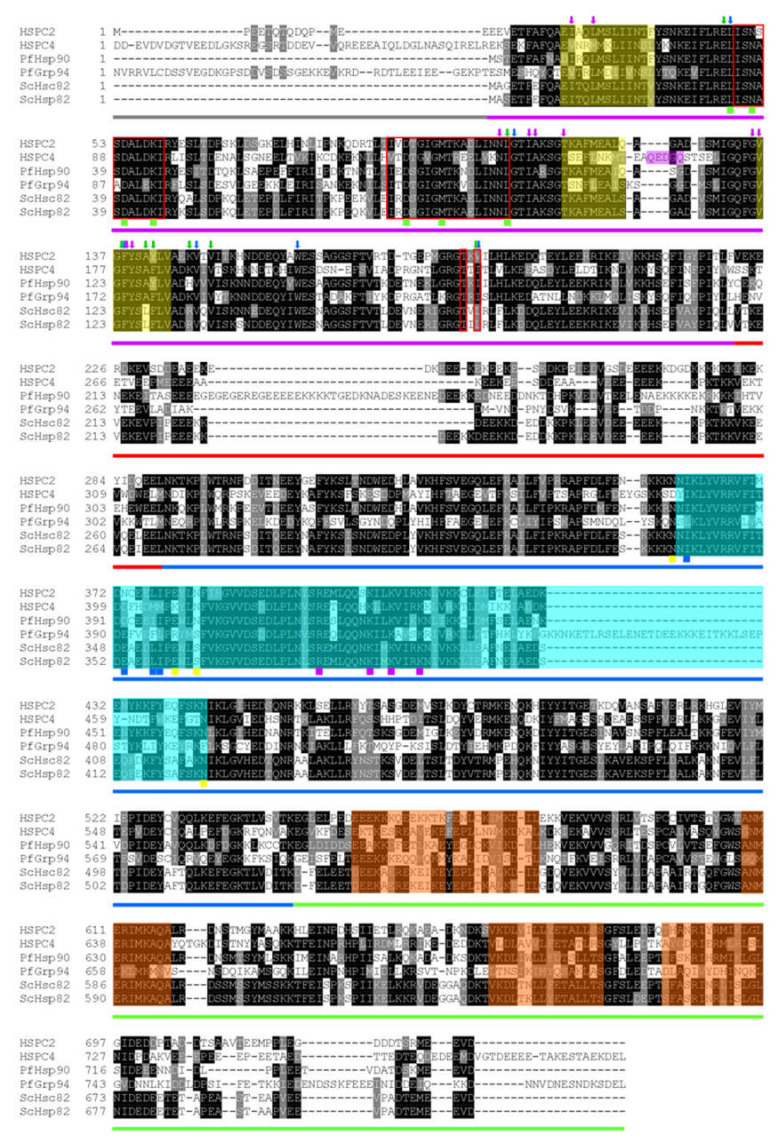
Multiple sequence alignments of Hsp90 proteins. Multiple sequence alignments of Hsp90 proteins from host *Homo sapiens* with sequences retrieved from NCBI (https://www.ncbi.nlm.nih.gov/protein, accessed on 13 April 2021) with respective accession number HSPC2 (NP_001017963.2) and HSPC4 (AAH66656.1). The *P. falciparum* Hsp90 protein sequences were retrieved from PlasmoDB (www.plasmoDB.org, accessed on 13 April 2021) with accession numbers PfHsp90 (PF3D7_0708400) and PfGrp94 (PF3D7_1222300), respectively. The model eukaryote *Saccharomyces cerevisiae* Hsp90 protein sequences were retrieved from https://www.yeastgenome.org/(accessed on 13 April 2021) ScHsc82 (accession numbers: S000004798) and ScHsp82 (Accession number: S000006161). Residues involved in ATP binding are indicated in red boxes [30,31]. The arrowheads represent hydrophobic side pocket one in blue, side pocket two in green, and side pocket three in purple. The QEDGQ insertion of human Grp94 is highlighted in purple. The lid helixes N1, N4, and N5, are highlighted in yellow. The residues that take part in the binding of geldanamycin in the NTD are indicated with green squares. The putative substrate-binding pocket of the MD is shown in cyan. Critical hydrophobic residues in this cavity are indicated with blue squares. Residues involved in hydrogen bonding with gambogic acid are indicated with yellow squares. Critical catalytic loop residues of the MD involved in ATP hydrolysis are indicated with purple squares. Colored bars beneath the MSA indicates the protein domains: Grey is the signal peptide and Pre-N region. The NTDs are indicated with purple bars, the linker with red, the MDs with blue, and the CTDs with green. The conserved residues are shown with white text on black background, and semi-conserved residues are shown with white text on a grey background. The protein sequences were submitted to T-Coffee (http://tcoffee.crg.cat/apps/tcoffee/do:regular; accessed on 14 September 2021)) for alignment. Boxshade (http://www.ch.embnet.org/software/BOX_form.html; accessed on 14 September 2021) conservation level coloring.

**Figure 3 cells-10-02849-f003:**
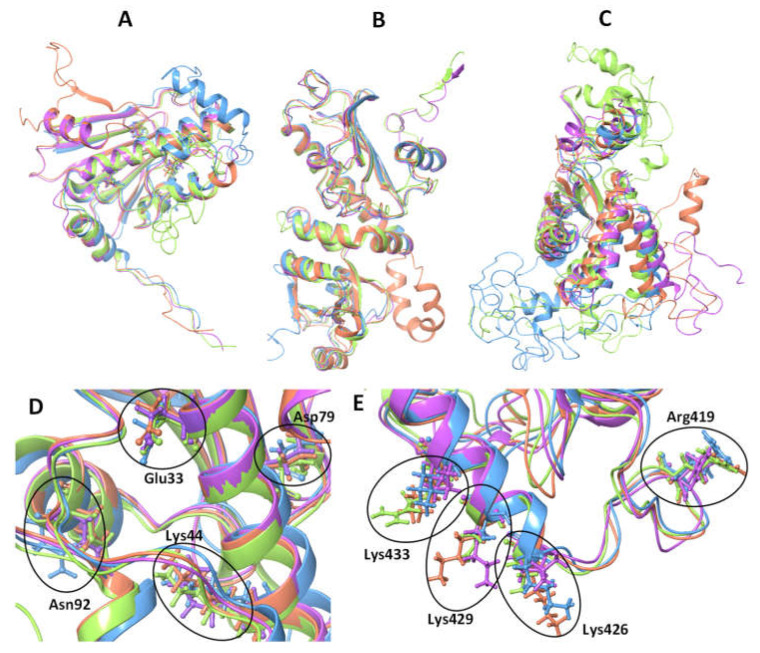
Superimposed domains and catalytic loop residues of the four parasite Hsp90 proteins. The 3-dimensional models of the four parasite Hsp90 proteins are superimposed to show catalytic residue structural conformation conservation. Catalytic residues of the other Hsp90 paralogs are thought to interact with the γ phosphate of ATP, enabling its hydrolysis. (**A**) The NTDs of the four parasite proteins. (**B**) The MDs of the four parasite proteins. (**C**) The CTDs of the four parasite proteins. The four parasite proteins superimposed cartoon zoomed-in for the (**D**) MD and (**E**) the NTD. The catalytic residues of PfHsp90 are labeled in (**D**,**E**). PfHsp90 is shown in purple, PfGrp94 in orange, PfHsp90_A in blue, and PfHsp90_M is shown in green. The non-conserved residues are in the MD of PfHsp90_M, which are Ser513, Asn516, and Arg520. Protein structures 3-D models were created by depositing amino acid sequences on the PHYRE2 protein fold recognition server (http://www.sbg.bio.ic.ac.uk/phyre2/html/page.cgi?id=index; accessed on 18 April 2021). The resultant 3-D protein structures were visualized and superimposed using the Schrödinger Maestro Release 2021-3: LLC, New York, NY.

**Figure 4 cells-10-02849-f004:**
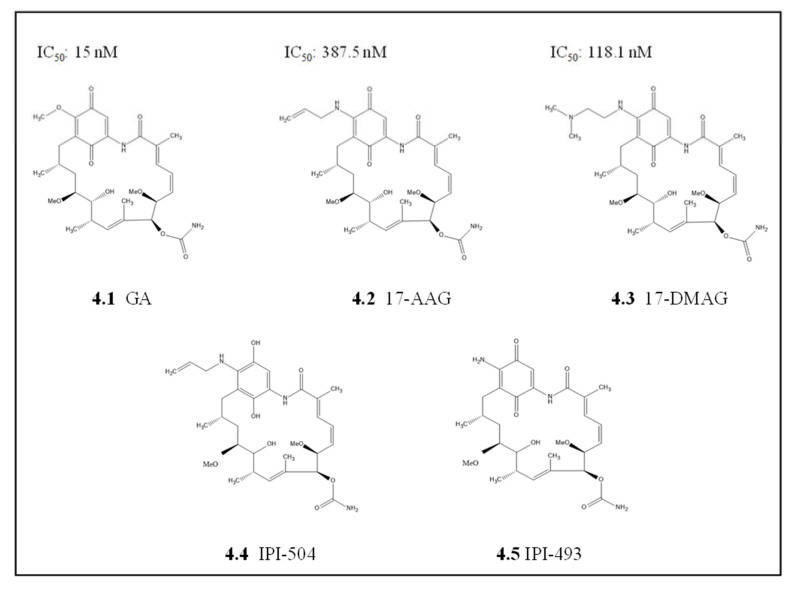
Chemical structure of geldanamycin and its derivatives. **4.1** Geldanamycin (GA) is a natural Hsp90 inhibitor and its derivatives **4.2** 17-allylamino-17-demethoxygeldanamycin (17-AAG), **4.3** 17-dimethylamino ethylamino-17-demethoxygeldanamycin (17-DMAG), and **4.4** 17-allylamino-17-demethoxygeldanamycin hydroquinone hydrochloride (IPI-504). **4.5** 17-amino-17-demethoxygeld anamycin (IPI-493) is a major metabolite of both IPI-504 and 17-AAG. The *P. falciparum* growth inhibition IC50 values are indicated.

**Figure 5 cells-10-02849-f005:**
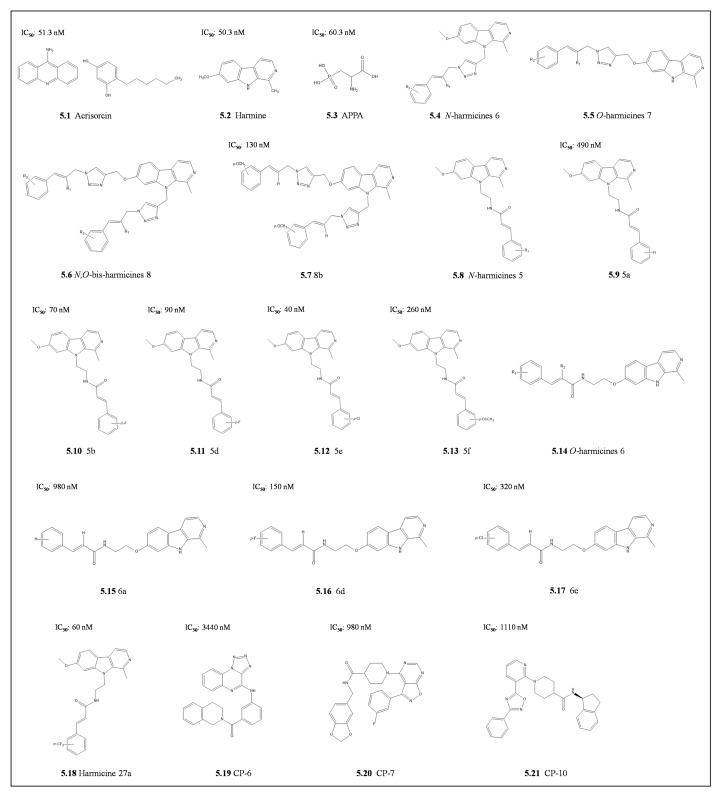
The chemical structure of Harmine and the derivatives. The inhibitors, **5.1** Acrisorcin. **5.2** Harmine. **5.3** 2-amino-3-phospho propionic acid (APPA) were tested in in parsite growth inhibition studies. The **5.4** *N*-harmicines 6 and **5.5** *O*-harmicines 7. **5.6** *N,O*-bis-harmicines 8 and their derivative **5.7** 8b. **5.8** *N*-harmicines 5 and its derivatives **5.9** 5a, **5.10** 5b, **5.11** 5d, **5.12** 5e and **5.13** 5f. **5.14** *O*-harmicines 6 and its derivatives **5.15** 6a, **5.16** 6d and **5.17** 6e. **5.18** Harmicine 27a. Other synthetic compounds **5.19** CP-6. **5.20** CP-7. **5.21** CP-10 were also developed. The respective IC_50_ values for parasite growth inhibition are shown.

**Figure 6 cells-10-02849-f006:**
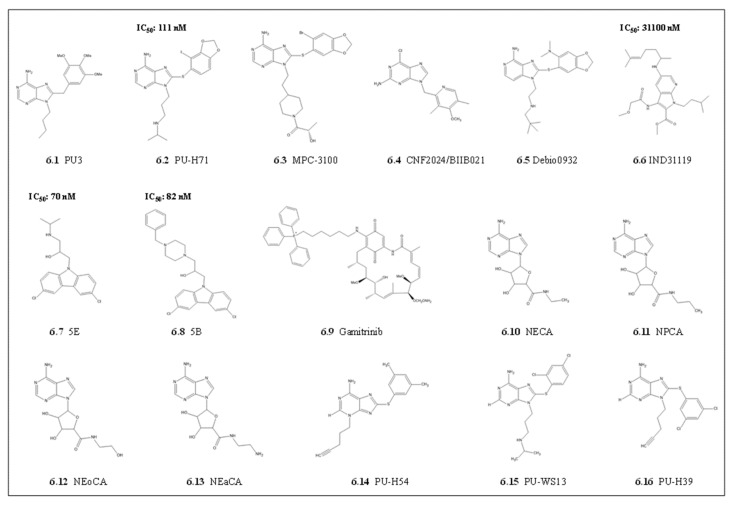
The purines and other derivatives as potential *P. falciparum* Hsp90 inhibitors. The chemical structures of the compounds are shown with their respective numbering. **6.1** PU3. **6.2** PU-H71. **6.3** MPC-3100. **6.4** CNF2024/BIIB021. **6.5** Debio0932. **6.6** Methyl 1-isopentyl-3-(2-methoxyacetamido)-5-((6-methylhept-5-en-2-yl)amino)-1H-pyrrolo[2,3-b]pyridine-2-carboxylate (IND31119). Two amino-alcohol-carbazole (N-CBZ) **6.7** 5E and **6.8** 5B. **6.9** Gamitrinib. **6.10** N-ethyl-carboximido adenosine (NECA) and its derivatives **6.11** N-propylcarboxamido adenosine (NPCA), **6.12** N-hydroxyethylcarboxamido adenosine (NEoCA) and **6.13** N-aminoethycarboxamido adenosine (NEaCA). Three purine derivatives **6.14** 8-((2,4-Dimethylphenyl)thio)-3-(pent-4-yn-1-yl)-3H-purin-6-amine (PU-H54), **6.15** 6-amino-8-[(3,5-dichlorophenyl)thio]-N-(1-methylethyl)-9H-purine-9-propanamine (PU-WS13) and **6.16** 8-((2,4-Dichlorophenyl)thio)-9-(pent-4-yn-1-yl)-9H-purin-6-amine (PU-H39). Known parasite growth inhibition IC_50_ values are indicated. purine and derivatives, 7 a-azandole and amino-alcohol, carbazole derivatives with known parasite growth inhibition IC_50_ values, are indicated.

**Figure 7 cells-10-02849-f007:**
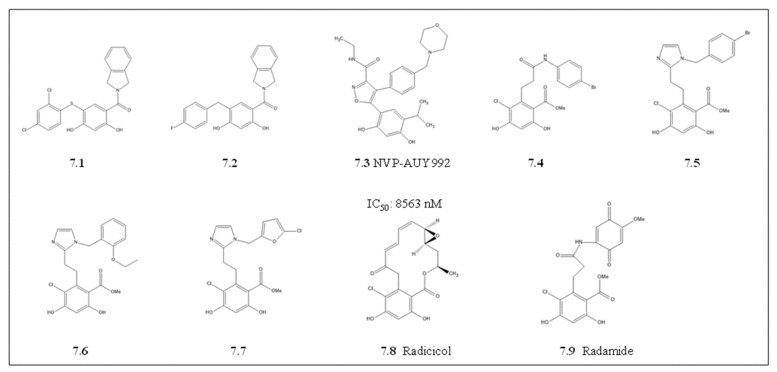
Radicicol and derivatives. The chemical structure of radicicol (**7.8**) derivatives include Radamide (**7.9**) and synthetic derivatives, compounds **7.1** and **7.2**.NVP-AUY922 (**7.3**) is a resorcinylic isoxazole amine compound, with its derivatives **7.4**, **7.5, 7.6** and **7.7** The respective IC_50_ value for parasite growth inhibition is shown.

**Figure 8 cells-10-02849-f008:**
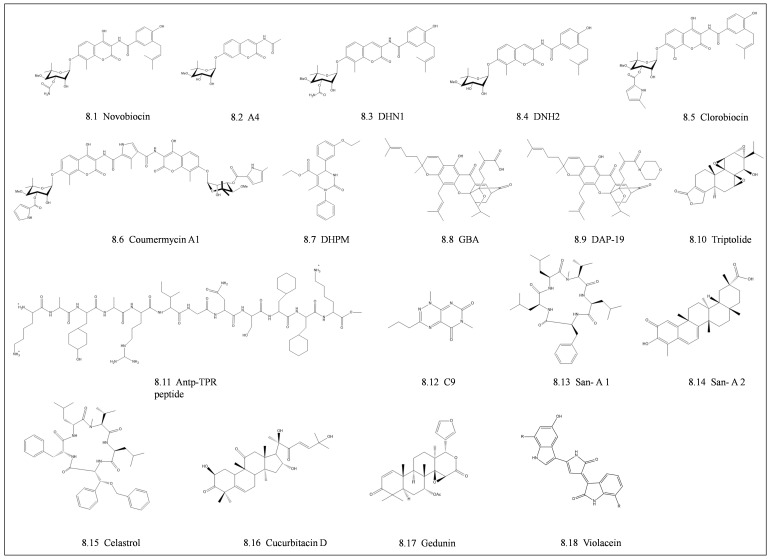
The chemical structure of inhibitors targeting the Hsp90 middle and C-terminal domains. The structures of compounds that target the NTD and CTD of Hsp90 and also abrogate co-chaperone interactions are shown. Inhibitors of the coumermycin family **8.1** Novobiocin and its derivatives **8.2** A4, **8.3** 4-deshydroxynovobiocin (DHN1) and **8.4** 3′-descarbamoyl-4-deshydroxynovobiocin (DHN2)**.** Additional members of the coumermycin family **8.5** Clorobiocin and **8.6** Coumermycin A1 derivative. **8.7** 3,4-dihydropyrimidin-2(1H)-one (DHPM). A natural product **8.8** Gambogic acid (GBA) and its derivative **8.9** DAP-19. A diterpene triepozide **8.10** Triptolide (TL). A peptidomimetic compound **8.11** Antp-TPR peptide. A small molecule compound **8.12** 1,6-dimethyl-1-3-propylpyrimido(5,4-e)(1,2,4)triazine-5,7-dione (C9). Sansalvamide A compounds **8.13** San-A1 and **8.14** San-A2. Additional co-chaperone inhibitors **8.15** Celastrol, **8.16** Cucurbitacin D and 8.17 Gedunin. 8.18 Violacein.

**Table 1 cells-10-02849-t001:** Hsp90 proteins from *Plasmodium falciparum* and *Homo sapiens*.

Human Co-Chaperone	*P. falciparum* Homolog	Function	References
Hop	PfHop (PF3D7_1434300)	Adaptor for Hsp70 and Hsp90; inhibitor of ATPase function	[44,45,50,51]
Cdc37		Kinase-specific co-chaperone	[52]
TTC4		Interaction with Cpr7/Cyp40	[53]
Tah1	PfRPAP3/PfTah1 (PF3D7_0213500)	Component of Rvb1-Rvb2-Tah1-Pih1 (R2TP) complex	[54]
Pih1	PfPih1 (PF3D7_1235000)	Component of (R2TP) complex	[37,54,55]
Cyp40	PF3D7_1111800	Peptidylprolyl-cis/trans-isomerase	[48]
P23	Pfp23A (PF3D7_1453700) Pf23B (PF3D7_0927000)	Inhibits ATPase	[56,57]
Aha1	PfAha1 (PF3D7_0306200)	ATPase activator	[48,58]
	PfAha1 (PF3D7_1334200)	ATPase activator	
PP5	PfPP5 (PF3D7_1355500)	Phosphatase	[59,60]
Sgt1	PfCBP (PF3D7_0933200)	kinetochore assembly	[61]
FKBP38	PfFKBP35 (PF3D7_1247400)	Peptidylprolyl-cis/trans-isomerase	[62,63]
	PfCns1 (PF3D7_1108900)		[48]

## Data Availability

All Data is available in this manuscript.

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
