# Peer review of "Inhibitors of the Plasmodium falciparum Hsp90 towards Selective Antimalarial Drug Design: The Past, Present and Future"

_cells, 2021, doi:10.3390/cells10112849_

Round 1

Reviewer 1 Report

Dear authors,

I have read through your manuscript. Great effort writing this review, which covers not only challenges associated with targeting Plasmodial hsps but also the human hsps for cancer therapy. 

My observations are mostly on grammar and formatting issues. (please check the attached file)

And also the figure 2 with alignment of sequences need to be changed. Six sequences should be used not 10, to allow readers to read it easily. 

Author Response

Title:

Line 2: Plasmodium falciparum should be in italics, and throughout the manuscript. As well as all other biological names.

Author’s response: We have corrected all biological names in the manuscript

Abstract:

Line 11: first line treatment, artesunate.

Author’s response: We have corrected the the sentence and inserted the word ‘treatment’.

Line 12: to reverse antimalarial drug resistance development.

Author’s response: We appreciate the reviewer’s comment and we have corrected the sentence by adding the word ‘antimalarial’.

Introduction

Lines 19-21: obviously a lot of grammar will need to be corrected in this manuscript. Please correct here and throughout manuscript. For example, ‘Malaria still poses major public health burden globally, with an estimated 229 million cases and more than 409 000 deaths; most of these deaths being in the tropics [1].’ Authors need to put commas and periods where appropriate.

Author’s response: We have corrected the sentence now reads: ‘Malaria still poses major public health burden globally with an estimated 229 million cases and more than 409 000 deaths; most of these deaths  being in the tropics in the year 2019 [1]’.

Line 27: Beginning of a sentence, Plasmodium in full and Anopheles’ not anopheles’

Author’s response: We have corrected the sentence and wrote Plasmodium in full and upper case for Anopheles.

Line 34: proportion is better than a portion.

Author’s response: We have corrected the word and inserted proportion in place of portion.

Line 50 is not thought to be. It is known. Please revised.

Author’s response: We have corrected the sentence and replaced thought with ‘known’.

Line 51: what genes? Authors should provide some examples of some of the genes.

Author’s response: We have appreciate the reviewer’s comments and we have mentioned the gene ‘chloroquine resistance transporter’

Line 52-53: hsps mediate not only proper folding of mutated proteins. All kind of proteins even at basal level. Authors should remove mutated and focus on proper folding of proteins, especially during stress (chemical, heat, etc).

Author’s response: We have reworded our sentence now reads ‘These proteins Hsps facilitate the native folding of virtually all proteins under stress including drug pressure.’ However since the section is on drug resistance focus was based on drug resistance development. The other stresses are dealt with in later sections.

Lines 56-57: authors should provide reference/s to substantiate this claim.

Author’s response: We have addressed the issue by toning down on the claim and added a reference [10]

Lines 66-67: Due to the central role they…

Author’s response: We have corrected the sentence construction.

Lines 67-68: infectious diseases, such as malaria [14].

Author’s response: We have corrected the sentence punctuation.

Line 117: Figure 2: fonts too small. Too much information compacted makes it impossible to read properly. I suggest authors to reduce the number of these sequences to only 6, using only two sequences from humans, two from S. cerevisiae and maximum of 2 from Plasmodium falciparum. Authors can provide another figure in supplementary.

Author’s response: We appreciate the reviewer’s comments and we have redrawn Figure 2 to maximize text sizes for 6 proteins sequences and added a Supplementary Figure S1 with the full alignments.

Line 139: correct ‘encodes for four Hsp90…’

Author’s response: We have corrected the sentence to read ‘The P. falciparum genome encodes for four Hsp90 genes’.

Line 142: ER in full.

Author’s response: We have corrected ER to be in full Endoplasmic reticulum.

Line 215: Format this line which lacks a beginning.

Author’s response: We have corrected the table legend to read: ‘The co-chaperones gene names with PlasmoDB accession numbers and functions are shown with. Tt he abbreviated protein names are as follows:’

Line 249: Plasmodium in full.

Author’s response: We have used the full plasmodium falciparum in the subheading.

Line 293: please remove ‘research’

Author’s response: We have deleted the word ‘research’ throughout the manuscript.

Line 341: remove extra space after ‘segment’

Author’s response: We removed the extra space.

Line 365: factors that factors that. Please correct.

Author’s response: We have deleted the repeated words.

Line 783; References need to be formatted. Double numbering on the lists and improper formatting issue.

Author’s response: We have deleted double numbering and re-formatted the references.

Reviewer 2 Report

This is a good review that describes the work that has been published so far on malaria Hsp90. I found it informative and it will be a useful guide to those interested in this field. However, it’s important to say that a book chapter addressing this subject was recently released. I noticed some relatively small errors which I mention below and have also made some small suggestions.

  1. Line 29. The sporozoites are injected primarily into the dermis, not into the blood stream;
  2. Line 37. The symptoms associated to malaria are not just due to release of parasitic and iRBC intracellular material. For example, adhesion phenomena in falciparum malaria is associated with disease severity;
  3. Line 42. I think it might be good to specify the observation of increased clearance times with ACTs rather than just calling it resistance;
  4. Line 50. The development of drug resistance is not just due to parasite targeted gene mutations. It can also occur due to mutation in genes related to the influx/efflux pumps that affect intraparasitic concentrations of the drug (crt, for example);
  5. Lines 54-57. I am confused how the spatial organization of hsp90 and crt genes could promote drug resistance. This needs a little clarification;
  6. Item 1. Here, I missed a brief explanation of the role of Hsp90 during heat stress on Plasmodium development (see the references doi: 10.1074/jbc.M409165200 and 10.1038/s41564-021-00940-w);
  7. Figures 1 and 3. I did not find whether the protein alignments were performed using known crystal structures or using models. In case models were used, please provide a brief explanation of how the models were built;
  8. Line 364. I am not sure if all 4 Hsp90 are essential for parasite development, since the 4 members of the parasite Hsp90 family have not been studied separately (at least in the references mentioned). In fact, here's my biggest concern with the study of Hsp90 in malaria parasites. The vast majority of studies use known human Hsp90 inhibitors to determine the function and essentiality of parasite Hsp90. How can it be ensured that the effect of such inhibitors on parasite development is specific to Hsp90 and not due to other targets? The authors could consider mentioning this;
  9. Line 365. “that factors” seems to be a typing error;
  10. Lines 444-454. I found it interesting to note that the binding affinity of the compound can vary according to the protein used in enzymatic assay.

Arguably, the review could benefit from a short section, describing briefly the types of enzymatic assays used to test the Hsp90 inhibition of malaria parasites;

  1. Line 483. I believe that Ref 10 is a Review article and so it may not be appropriate here;
  2. Lines 483 and 551. Please confirm if chemosensitization of chloroquine-resistant parasites in the presence of Hsp90 inhibitors has been evaluated. To say that a compound is synergistic with chloroquine is not to say that it reverses chloroquine resistance.
  3. I believe it would be informative to add PfHsp90 enzymatic and antiplasmodial activity, where available, to the structures of the article.

Author Response

This is a good review that describes the work that has been published so far on malaria Hsp90. I found it informative and it will be a useful guide to those interested in this field. However, it’s important to say that a book chapter addressing this subject was recently released. I noticed some relatively small errors which I mention below and have also made some small suggestions.

Author’s response: We have updated our references and cited the new book on Plasmodial Hsp90s as reference [29]

  1. Line 29. The sporozoites are injected primarily into the dermis, not into the blood stream;

Author’s response: We have corrected our sentence to read: …injects the sporozoites into the dermis and find their way to the blood stream’.

  1. Line 37. The symptoms associated to malaria are not just due to release of parasitic and iRBC intracellular material. For example, adhesion phenomena in falciparum malaria is associated with disease severity;

Author’s response: We have appreciate the reviewer’s comments and reworded our sentence to read: ‘However, the erythrocytic cycle of the parasite life cycle is mostly associated with malaria symptoms, for example, the adhesion phenomenon is associated with disease severity in P. falciparum’.

  1. Line 42. I think it might be good to specify the observation of increased clearance times with ACTs rather than just calling it resistance;

Author’s response: We have corrected our statement to read: ‘However, its efficacy is under threat due to the increased clearance times observed with ACTs which are associated with the development of resistance to Artemisinin monotherapy which has been reported in the Great Mekong region and recently in Africa’.

  1. Line 50. The development of drug resistance is not just due to parasite targeted gene mutations. It can also occur due to mutation in genes related to the influx/efflux pumps that affect intraparasitic concentrations of the drug (crt, for example);

Author’s response: We have corrected the sentence to: ‘The proper functioning of such mutated genes related to the influx and efflux pumps affect intraparasitic concentrations of the drug for example, the correct folding of  mutated chloroquine resistance transporter (crt)  is attributed to molecular chaperones such as the heat shock proteins (Hsp)’.

  1. Lines 54-57. I am confused how the spatial organization of hsp90and crt genes could promote drug resistance. This needs a little clarification;

Author’s response: We have corrected our sentence and toned down on the potential association of gene clusters to read: ‘This suggest that the regulation through cis-regulatory elements such as transcription factors may similarly co-regulate the expression of both chaperone and the mutated crt gene. In addition, some molecular chaperones have been shown to directly and indirectly interact with crt through a yet to be established mechanism [10]. It is tempting to speculate that the chaperoning role of Hsps on proteins may promote drug resistance during stress response’.

  1. Item 1. Here, I missed a brief explanation of the role of Hsp90 during heat stress on Plasmodiumdevelopment (see the references doi: 10.1074/jbc.M409165200 and 10.1038/s41564-021-00940-w);

Author’s response: We have added the following sentence: ‘The complex life cycle that the parasite survives under a stressful environment as it transitions from the poikilothermic vector at 25 °C to the homoeothermic host at 37°C and further increased in fever episodes to 41 °C resulting in thermal stress [7].. In order to sur-vive these, the parasite needs a robust protein quality control system, and the parasite up-regulates the expression of some of the molecular chaperones to maintain its proteome [8]’.

  1. Figures 1 and 3. I did not find whether the protein alignments were performed using known crystal structures or using models. In case models were used, please provide a brief explanation of how the models were built;

Author’s response: We have added the following brief explanation to the figure legends: Three-dimension (3-D) protein structures were generated by depositing protein sequences to the online with the PHYRE2 protein fold recognition server (http://www.sbg.bio.ic.ac.uk/phyre2/html/page.cgi?id=index). The 3-D model structures were visually inspected for structural correctness using the Schrodinger Maestro release 2021-3: LLC, New York, NY.

  1. Line 364. I am not sure if all 4 Hsp90 are essential for parasite development, since the 4 members of the parasite Hsp90 family have not been studied separately (at least in the references mentioned). In fact, here's my biggest concern with the study of Hsp90 in malaria parasites. The vast majority of studies use known human Hsp90 inhibitors to determine the function and essentiality of parasite Hsp90. How can it be ensured that the effect of such inhibitors on parasite development is specific to Hsp90 and not due to other targets? The authors could consider mentioning this;

Author’s response: We appreciate the reviewer’s comments and we have added the following sentences: ‘Similar to other eukaryotes, in P. falciparum, as eukaryotes it is plausible that all 4 Hsp90s are essential for parasite survival and erythrocytic stage transitions [65,106]. However, it should be noted that most studies on parasite Hsp90s use known human Hsp90 inhibitors to determine function and essentiality of parasite Hsp90s. There is need for caution in interpreting such results as it has not been thoroughly validated that the effect of such inhibitors on parasite development is specific to Hsp90 and not due to other targets’.

  1. Line 365. “that factors” seems to be a typing error;

Author’s response: We have deleted the repeated words.

  1. Lines 444-454. I found it interesting to note that the binding affinity of the compound can vary according to the protein used in enzymatic assay.

Author’s response: We appreciate the reviewer’s comments and we have clarified that the variations observed were only in binding assays and not catalytic enzymatic assays. We removed the reference to catalytic activities from this section.

Arguably, the review could benefit from a short section, describing briefly the types of enzymatic assays used to test the Hsp90 inhibition of malaria parasites;

Author’s response: We appreciate the reviewer’s comments and there are limited enzymatic assays conducted on parasite Hsp90 as most reports focus on binding affinities. Therefore, we have added a sentence in section 4 as follows: ‘As such, the most common Hsp90 enzymatic activity assay employed is the ATPase assay in parasite Hsp90 inhibition studies [8,10,106]’.  

  1. Line 483. I believe that Ref 10 is a Review article and so it may not be appropriate here;

Author’s response: We have corrected the reference to a research article in place of the review.

  1. Lines 483 and 551. Please confirm if chemosensitization of chloroquine-resistant parasites in the presence of Hsp90 inhibitors has been evaluated. To say that a compound is synergistic with chloroquine is not to say that it reverses chloroquine resistance.

Author’s response: We have corrected the statement and deleted the chloroquine resistance reversal as there was no evidence for chemosensitization.

  1. I believe it would be informative to add PfHsp90 enzymatic and antiplasmodial activity, where available, to the structures of the article.

Author’s response: We have added the P. falciparum growth inhibition IC50 values to all the figures, where available.